# Extracting Latent Generalization from Models Trained with Noisy Labels

**Simran Ketha**[1,2] **& Venkatakrishnan Ramaswamy**[1,2]
[1]Department of Computer Science & Information Systems,
Birla Institute of Technology & Science Pilani, Hyderabad 500078, India
[2]Anuradha & Prashanth Palakurthi Centre for Artificial Intelligence Research,
Birla Institute of Technology & Science Pilani, Hyderabad 500078, India.
`{p20200021,venkat}@hyderabad.bits-pilani.ac.in`

## Abstract

Overparameterized Deep Neural Networks that generalize well underlie much of the recent successes in Deep Learning. It has also been known that when training data labels are noisy, Deep Networks, on training, exhibit the capacity to learn this label noise, which hurts their generalization, as manifested by degraded test accuracies. Here, we investigate whether we can extract more reliable predictions from these models whose predictive power is impacted by such unreliable training data, while sticking to the standard training paradigm. Specifically, we consider the question of extracting better generalization from the latent representations of the layers of the model, in this setting. To this end, we study the class-conditional subspaces corresponding to the training data corrupted with label noise. Furthermore, we examined the geometry of the layerwise outputs in relation to these subspaces. We find, surprisingly, that doing so leads to a technique to extract significantly better generalization than provided by the corresponding model. We show results exemplifying this phenomenon on multiple models trained with a number of standard datasets. Our work demonstrates that we can extract underutilized latent generalization present in the internal representations of models trained with data which has label noise.

## 1 Introduction

Much of modern Deep Learning utilizes overparameterized models which have been observed to be able to generalize remarkably well to unseen data. However, Deep Network models have also been shown to possess the ability to memorize training data. In particular, it has been shown [29, 30] that when one shuffles class labels of data points from standard training datasets to varying degrees, Deep Networks can still have high/perfect training accuracy on such corrupted training data; however, this appears to typically be accompanied by poor performance on unseen test data (that have true labels). This phenomenon has been called *memorization*, since it is thought that the model rote-learned the training data at the expense of acquiring the ability to generalize to unseen examples. In real-world settings, where training data has label noise, the phenomenon of memorization impacts the ability of models to reliably perform on unseen data.

Here, we study the organization of subspaces of class-conditional training data on layerwise outputs of a number of Deep Networks. We estimate these subspaces using Principal Components Analysis (PCA). To further understand this organization, we built a simple classifier that leverages the geometry of the layer output of an incoming datapoint, relative to these class-conditioned subspaces. Specifically, we measure the angle between this output vector and its projection on each of these

39th Conference on Neural Information Processing Systems (NeurIPS 2025) Workshop: Reliable ML from Unreliable Data.

class-conditioned subspaces. The classifier then predicts this datapoint's class to be the class whose subspace has the minimum such angle. We find, surprisingly that this classifier applied to certain layers, has significantly better generalization performance than the model itself. This is remarkable because it also indicates that the layerwise representations of the Deep Network retain significant latent ability to generalize, even in the face of such noisy training data, and that this ability can be extracted via a simple probe built only by using the corrupted training dataset.

## 2    Related work

The idea of probing intermediate layers of Deep Networks isn't new. For example, [16, 1] do so by using kernel PCA & linear classifiers respectively. However, this approach has not been used to investigate memorization. Indeed, [1] explicitly avoid examining memorized networks from [29] because they thought such probes would inevitably overfit. Our results are therefore especially surprising in this context.

There is evidence that DNN's learn simple patterns first, before memorizing [2], & DNNs learn lower frequencies first [3]. [24] study memorized models, concluding that memorization happens in later layers, since rewinding early layer weights to their early stopping values recovers some generalization, but rewinding later layer weights doesn't. On the contrary, our results suggest that later layers in most models investigated retain significant ability to generalize, & we demonstrate this without modifying the weights of the trained network.

There is an important line of theoretical work in deep linear models [21] where the question of generalization has been studied. In this context, [12] offer a theoretical explanation for the phenomenon of memorization in networks trained with noisy labels.

Experiments towards understanding training dynamics across layers using different Canonical Correlation Analysis have be explored [18] and in various generalized and memorized networks is analyzed [17]. Centered Kernel Alignment in different random initializations by [9] and network similarity between model trained with same data and different initialization is examined by [27]. Also experiments related to using measures of the representational geometry towards understanding dynamics of layerwise outputs [4, 5].

To deal with label noise, many heuristics have been explored [8, 22, 19, 31, 14] & for classification task see [7, 20, 15, 23]. For over parameterized models, [13] shows that the memorized network weights are far away from the initial random state in order for them to overfit the noisy labels. [25] propose a theoretical model for epochwise double descent that suggests that for small-sized models, moderate amounts of noise can cause generalization error to dip later on in training.

## 3    Methodology

To interpret and understand the organization of layerwise learned representations in memorized networks, we first construct class-conditioned subspaces corresponding to the corrupted training data.

**Creation of subspaces**: For a specific layer, we estimate subspaces for each class. The class-conditioned training data subspaces on layerwise outputs of Deep Networks are computed using PCA. If the empirical mean of the class-conditioned data isn't zero, PCA in effect, will provide us an affine space, i.e. a linear space that doesn't pass via the origin. However, we have determined subspaces – which are linear spaces passing through the origin – here rather than affine spaces. In order to do so, we add the negative of each sample to the dataset so it is guaranteed to have empirical mean be zero, before running PCA. This created dataset is sent to the PCA algorithm to calculate PCA components for a certain percentage of variance explained in the dataset. The span of these PCA components is the subspace $S$. We illustrate the process of creating subspaces for a Multi-layer Perceptron (MLP) model in Figure 1.

To studied the geometry of the layerwise outputs in relation to these subspaces, we build a Minimum Angle Subspace Classifier (MASC) with the following steps:

**Projection of the data point**: Layer output of an incoming data point is projected onto the class-specific subspaces.

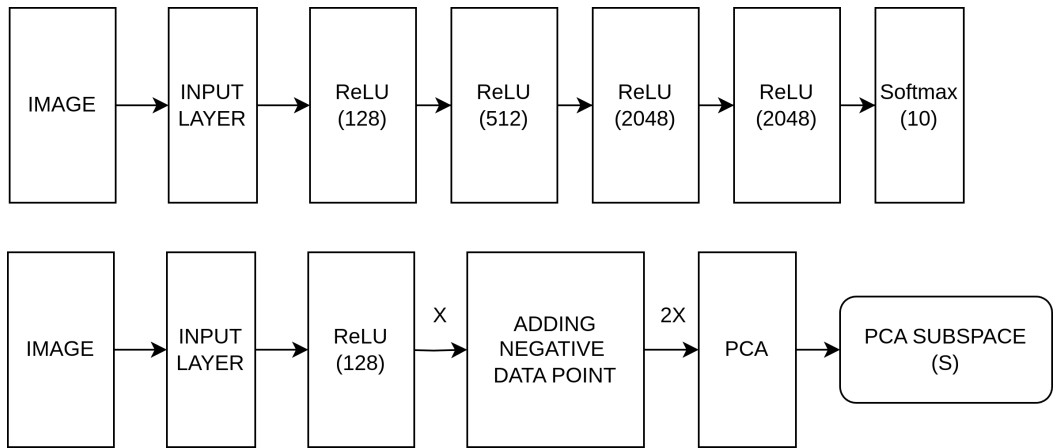

Figure 1: Class-conditioned training data subspaces on layerwise outputs of MLP using PCA. Top: Schematic of MLP model used in the work. Bottom: Creating the class-conditioned training subspace for ReLU (128) layer where 128 are the number of neurons.

**Label assignment using minimum angle**: For every data point, the angle between the original data point and projected data point for each class is calculated. The Minimum Angle Subspace Classifier (MASC) assigns to the datapoint, the label of the subspace having the minimum angle with the original data point.

While the subspaces are estimated using the training data alone, accuracy of the Minimum Angle Subspace Classifier is determined for the training data and the testing data separately. This process is followed for all the layers in the network independently. For experiments in Section 4, MASC uses corrupted training labels to create class-specific subspaces. We have used 99% as the percentage of variance explained, unless otherwise mentioned.

## 3.1 Experimental Setup

We have used multiple models and datasets, namely Multi-layer Perceptron (MLP) trained on MNIST [6] and CIFAR-10 [10] datasets, Convolutional Neural Networks (CNN) [1] trained on MNIST, Fashion-MNIST [28], and CIFAR-10 and AlexNet [11] trained on CIFAR-100 [10]. We have trained these models with training data having true labels ("generalized models") as well as separately using training data with labels shuffled to varing degrees ("memorized models") [30].

For memorized models, when we say we train it with corruption degree $p$, we mean that with probability $p$, we attempt changing the label for a training datapoint. Changing the labels happens uniformly at random. Note that this may result in the label remaining the same; therefore the expected fraction of datapoints whose labels changed are $p - p/c$ where $c$ is the number of classes. So, this would mean that for corruption degrees of 20% , 40%, 60%, 80%, 100% the expected percentage of training datapoints with changed labels is 18%, 36%, 54%, 72%, 90% respectively, when $c = 10$. We have run experiments for values of $p$ being 0% (generalized model), 20% , 40%, 60%, 80%, 100% (memorized models).

A summary of the models and datasets with training set size and number of parameters is in Table 1. The average training and testing accuracies of all the models over three runs are shown in Table 3 and 4 in section A.2. More details of these models, hyperparameters & training are available in Section A.1. Following standard practice in probing memorized models (e.g. [24]), we do not use explicit regularizers such as Dropout or batchnorm, or early stopping, as a result of which our baseline test accuracy numbers are often much lower than what is usually found with standard training of these models. All the models are trained to either reach very high training accuracies (i.e. $99\% - 100\%$) or trained until 500 epochs. Some models did not result in such high accuracies, in which case, results have been shown on the model obtained at epoch 500. We trained 3 instances of each model and

---

[1]The CNN models were built along the lines of [26].

Table 1: Training set size of the data sets and the number of parameters of the models.

| Model | Dataset | Training set size | Number of parameters |
|---|---|---|---|
| MLP | MNIST | 60,000 | 5,433,994 |
| | CIFAR-10 | 50,000 | 5,726,858 |
| CNN | MNIST | 60,000 | 344,042 |
| | Fashion MNIST | 60,000 | 344,042 |
| | CIFAR-10 | 50,000 | 456,330 |
| AlexNet | CIFAR-100 | 50,000 | 38,738,952 |

results displayed are averaged over these instances with the shaded region indicating the range of results also indicated in the plots.

Once the model is trained, we apply MASC on each layer of the network with respect to different subspaces. For MLP models, MASC experiments were performed for all the layers in the network including on the input (after it is pre-processed). For CNN models and AlexNet model, the experiments were performed on flatten layer (Flat) and fully connected layers (FC). While we ran the experiments on the input layer for CNNs, we did not do so for AlexNet.

## 3.2 Terminology

The general terminology used in this work is as follows:

- **Model Training Accuracy**: The model accuracy on the training set with corrupted labels.

- **Model Testing Accuracy**: The model accuracy on the testing data set with true labels.

- **Minimum Angle Subspace Classifier (MASC) Accuracy on Corrupted Training**: Training accuracy of MASC on training data set with respect to corrupted labels

- **Minimum Angle Subspace Classifier (MASC) Accuracy on Original Training**: Training accuracy of MASC on training data with respect to true training labels.

- **Minimum Angle Subspace Classifier (MASC) Accuracy on Testing**: Testing accuracy of MASC on testing data set with true labels was used.

## 4 Results

Models trained with corrupted labels have high training accuracy (on corrupted labels) while also having low testing accuracy [30]. We sought to leverage the class-conditioned subspaces of the hidden layers of these memorized models to obtain better generalization.

To do so, we build a Minimum Angle Subspace Classifier (MASC) using class-conditioned corrupted training subspaces obtained from the memorized models' hidden layer outputs. MASC is performed layer-wise for all the layers of the network independently as described in Section 3. MASC accuracy on corrupted training data, MASC accuracy on original training data, and MASC accuracy on testing data over the layer of MLP trained on MNIST, CNN trained on Fashion-MNIST, AlexNet trained on CIFAR-100 are shown in Figure 2. Experiments on MLP trained on CIFAR10, CNN trained on MNIST and CIFAR10 are shown in Figure 3.

Importantly, for every corrupted model we have (with non-zero corruption degree), except those with 100% corruption degree, we find that our Minimum Angle Subspace Classifier (MASC) in at least one layer has better testing accuracy than the corresponding model itself. In many cases, the MASC testing accuracy is dramatically better than that of the model. This is remarkable, because, in addition to the layerwise outputs, the MASC used precisely the same information (including the same corrupted training dataset) that was available to the model itself, and yet is able to extract better generalization. This suggests that the model retains significant latent generalization ability, which is not captured in its own test-set performance. Below, we make more specific observations on the performance of the models.

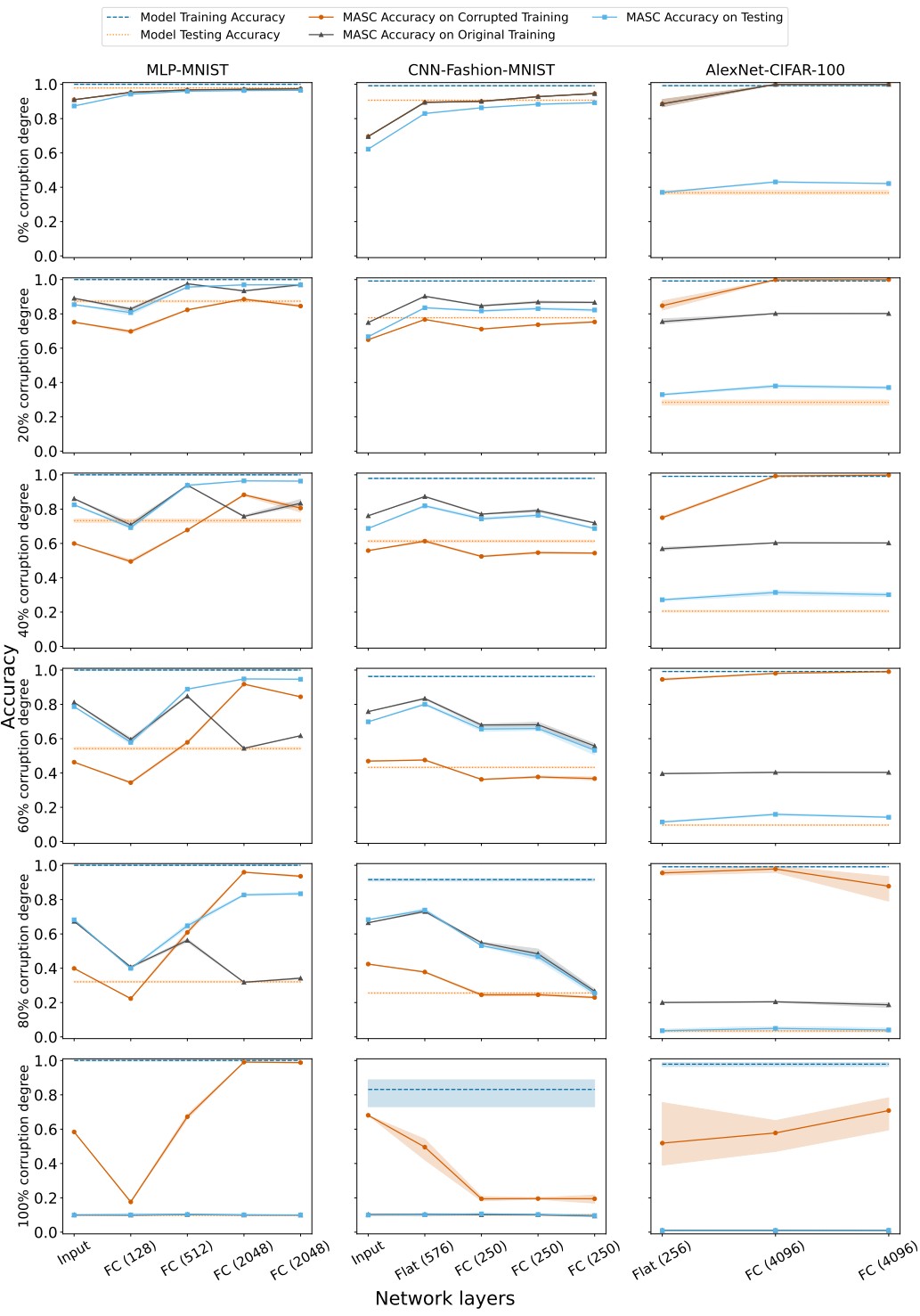

Figure 2: Minimum Angle Subspace Classifier (MASC) accuracy over the layers of the network when the data is projected onto corrupted training subspaces with the indicated corruption degree, for multiple models/datasets. Rows corresponds to plots with the same corruption degree & the columns correspond to the models, as noted. Training accuracy (dashed line) & testing accuracy (dotted line) of the model is shown. FC corresponds to fully connected layer with $ReLU$ activation whereas Flat corresponds to flatten layer without $ReLU$ activation.

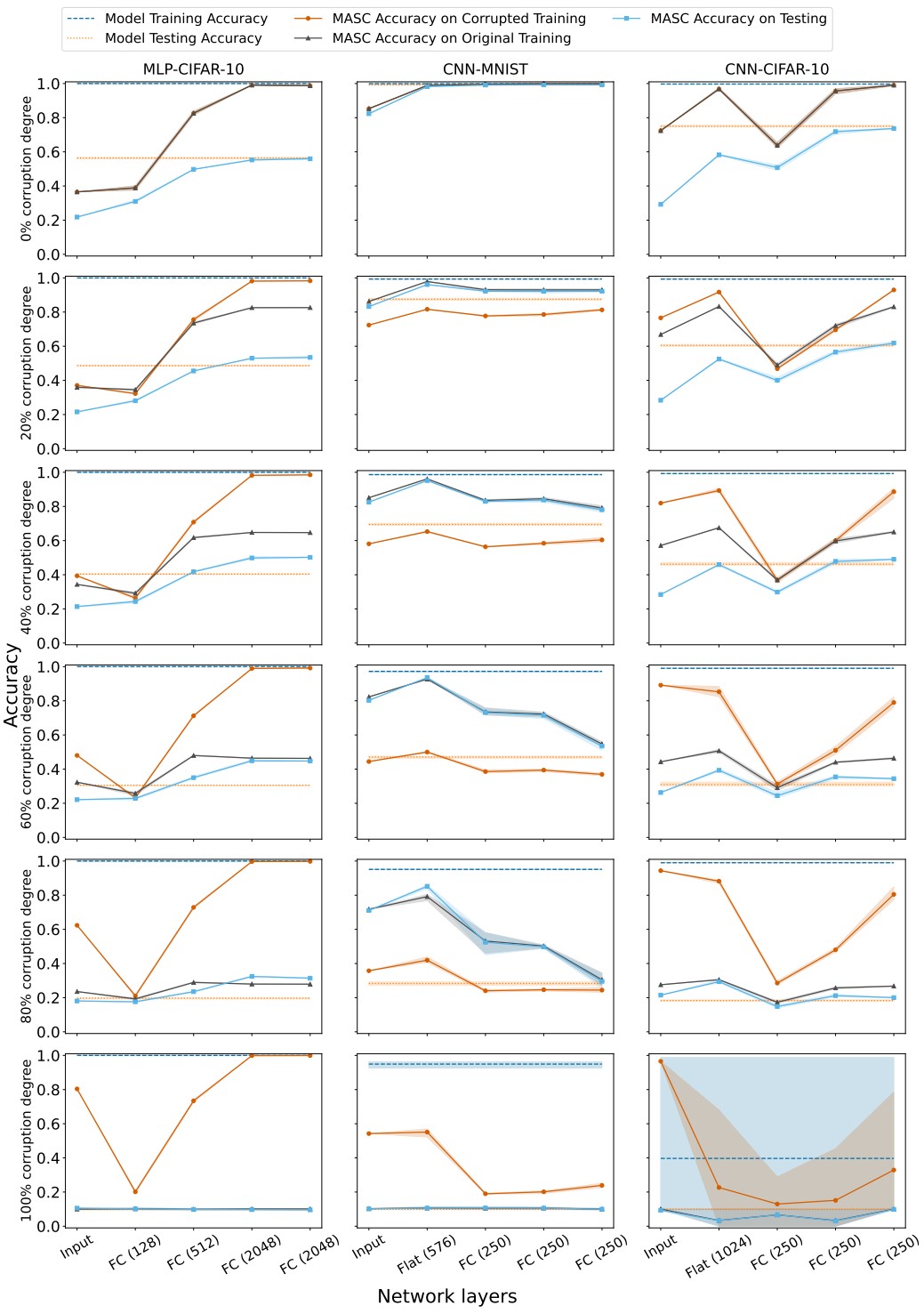

Figure 3: Minimum Angle Subspace Classifier (MASC) accuracy over the layers of the network when the data is projected onto corrupted training subspaces with the indicated corruption degree, for multiple models/datasets. Rows corresponds to plots with the same corruption degree & the columns correspond to the models, as noted. Training accuracy (dashed line) & testing accuracy (dotted line) of the model is shown. FC corresponds to fully connected layer with $ReLU$ activation whereas Flat corresponds to flatten layer without $ReLU$ activation.

With generalized models i.e. those with 0% corruption degree, at the later layers of the network, it is observed that in most of the cases MASC accuracy on training data approaches the models training accuracy. Similarly, MASC accuracy on testing data is comparable to or performed better than the models' test accuracy.

Even for high corruption degrees, we find that the MASC performs well. For example, with 80% corruption degree, which implies that approximately 72% of the training labels have been changed, we observed good MASC testing accuracy in many cases. Notably, the MASC test accuracy on the later layers is over 80% on MLP-MNIST, in comparison to 34% test accuracy by the model. Similarly, MASC test accuracy on one of the layers is about 75%, 4.9%, 30%, 80%, 28% for CNN-Fashion-MNIST, AlexNet-CIFAR-100, MLP-CIFAR10, CNN-MNIST and CNN-CIFAR10, in contrast to 25%, 3.4%, 20%, 29% and 19% model test accuracies respectively. In Table 2, for each degree of corruption and model-dataset, we also specifically list by what percentage the MASC classifier (for the best layer of the model) outperformed the models test accuracy.

Not only does the MASC have better accuracy than the model on the test data but it also does well on the training data with the true labels. Although the model has memorized the training data with corrupted labels, outputs from certain layers have the ability to predict the trained true labels. For example, in MLP-MNIST, for low to moderate degrees of corruption, MASC on the middle layer (FC (512)) has good accuracy on the true training labels, while also retaining good accuracy on the test set. With 40% corruption degree, approximately 36% are changed labels and yet the model has good accuracy on the true training labels in at least one layer of the network. e.g. MLP-MNIST has over 90% true training accuracy at layer FC(512), CNN-Fashion-MNIST has approximately 85% in Flat (576) layer, AlexNet-CIFAR-100 has approximately 60% in FC (4096) layer, MLP-CIFAR-10 has approximately 60% in FC (2048), CNN-MNIST has approximately 95% in Flat (576) & CNN-CIFAR10 has approximately 65% in Flat (1024). This means that almost 20% of those labels are predicted correctly even though the model was trained for 500 epochs or has reached high training accuracy on corrupted labels. In the process of doing this, the model does not have any direct information about the true labels and neither does the MASC.

Table 2: Percentage by which the MASC classifier (run on the best layer) outperformed the model's test accuracy. The accuracies in each case are averaged over three runs.

| Corruption degree | 20% | 40% | 60% | 80% |
|---|---|---|---|---|
| MLP-MNIST | 10.92% | 31.63% | 75.04% | 159.93% |
| MLP-CIFAR-10 | 9.89% | 24.41% | 46.97% | 64.74% |
| CNN-MNIST | 9.81% | 37.03% | 98.69% | 201.05% |
| CNN-Fashion-MNIST | 7.48% | 33.49% | 84.92% | 188.99% |
| CNN-CIFAR-10 | 2.29% | 6.26% | 27.03% | 60.17% |
| AlexNet-CIFAR-100 | 33.57% | 53.10% | 64.85% | 44.99% |

One way to think about a Deep Network, is as one that successively transforms input representations in a manner that aids in good prediction performance. Therefore, performance of the MASC on the input is a good baseline measure to assess if subsequent layers have favorable accuracies. Naïvely, for models trained with corrupted data, one would expect layered representations that enable the model to do well on the corrupted training data, but not do well on the test data or the training data that have true labels. While this expectation seems to hold with respect to the model itself, we find that the layer-wise representations do not necessarily follow this expectation. That is, MASC applied to subsequent layers, often have better true training accuracy and test accuracy than the MASC applied to the input, suggesting that the Deep Network does indeed transform the data in a manner more amenable to correct prediction, even if its labels are dominated by noise.

## 5  Discussion

In this work, we study models in the setting where they are trained with label noise. Such models are known to fit the training label noise very well, which comes at the cost of degraded generalization performance. We sought to leverage internal representations of such trained models in order to extract better generalization than that manifested by the models. Specifically, we were interested in studying

the geometry of the internals of the network, to this end. We constructed subspaces of the ambient space of layerwise outputs of the network, corresponding to individual classes, as indicated in the (corrupted) training data. On examining the geometry of where unseen datapoints mapped to in this ambient space, relative to the aforementioned subspaces, we found surprisingly that they turned out to be closer to the correct class label, more often than what the model itself would predict. This led to the MASC classifier, which can be deployed on layerwise outputs of models trained on data with label noise in order to obtain better test accuracies than the model itself. We demonstrated that the MASC classifier works favorably on multiple models trained on a number of standard datasets.

An interesting question is about why this phenomenon even occurs; naïvely one would expect that networks, on being trained with highly noisy data, discard the ability to generalize in favor of learning noise. Are there specific inductive biases that promote such generalization? And, do such mechanisms also promote generalization in networks whose training data isn't corrupted significantly by such noise? It would also be instructive to study the dynamics of this form of generalization during training. It is known [2] that the model's test accuracy transiently peaks in the early epochs of training with corrupted data, before dropping while training accuracy of the corrupted training data rises. It is unclear whether this transient rise in model generalization is caused by the subspace organization seen here, and if so, why such subspace organization isn't degraded as much as the model's test error over further epochs of training.

In closing, we demonstrate that models trained in the label noise setting retain latent representations which allow for significantly improved generalization, which can be extracted without exorbitant computational overhead. This phenomenon can be leveraged to extract more reliable predictions from models trained on such unreliable data.

### Acknowledgments

Simran Ketha was supported by an APPCAIR Fellowship, from the Anuradha & Prashanth Palakurthi Centre for Artificial Intelligence Research. The work was supported in part by an Additional Competitive Research Grant from BITS to Venkatakrishnan Ramaswamy. The authors acknowledge the computing time provided on the High Performance Computing facility, Sharanga, at the Birla Institute of Technology and Science - Pilani, Hyderabad Campus. We thank Harsha Varun Marisetty for compute assistance for training some of the models used here.

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

# A Appendix

## A.1 Model Details

The MLP model has 4 hidden layers with 128, 512, 2048 and 2048 units respectively. $ReLU$ activation was used after every layer and for classification $softmax$ activation was applied. Learning rate of 1e-3 and momentum = 0.9 was used with $SGD$ optimizer was used. Batch size of 32 was used in all the models. Data set was normalized by dividing each pixel value with 255.

CNN network has 3 blocks, each consisting of two convolutional layers, one max pooling layer. These blocks are followed by three fully connected layers. Convolutional layers have 16, 32, and 64 filters, respectively with stride=1 and filter size = 3 × 3. Max pooling layer has stride of 1 and filter size of 2 × 2. The fully connected layers at the end has 250 units each. It was trained with $Adam$ optimizer with learning rate of 0.0002. For MNIST and Fashion-MNIST batch size of 32 whereas for CIFAR-10 batch size of 128 were used. Data set was normalized by subtracting the mean and diving by the standard deviation for each channel. $ReLU$ activation was used after every layer except pooling and $softmax$ activation for classification.

AlexNet model was slightly modified for the use of the dataset. Batch size of 128 and $Adam$ optimizer with learning rate of 0.0001 was used. CIFAR-100 dataset before training was normalized by subtracting the mean and diving by the standard deviation for each channel.

All experiments were conducted on servers and workstations equipped with NVIDIA GeForce RTX 3080, RTX 3090, Tesla V100, and Tesla A100 GPUs. The server environment was configured with Rocky Linux 8.10 (Green Obsidian), while the workstation operated on Ubuntu 20.04.3 LTS. Implementations were developed in Python using the PyTorch framework, with a fixed random seed (torch.manual_seed = 42) to ensure reproducibility. Memory usage varied depending on the specific models and experiments, and accuracy was employed as the primary evaluation metric for model performance. Most experiments completed within 12–24 hours for each run, except for AlexNet on CIFAR-100, which required more time.

## A.2 Training and testing performance of the models

Average training and testing accuracies of the models over three different runs used in this paper are shown in Tables 3 and 4.

Table 3: Average training accuracy in percentages of all the models over three runs over different corruption degrees (indicated in the last six columns).

| Model | Dataset | 0% | 20% | 40% | 60% | 80% | 100% |
|---|---|---|---|---|---|---|---|
| MLP | MNIST | 99.99 | 99.99 | 99.99 | 99.99 | 100 | 100 |
| | CIFAR-10 | 99.99 | 99.99 | 99.99 | 99.99 | 99.99 | 99.99 |
| CNN | MNIST | 99.90 | 99.32 | 98.62 | 97.25 | 95.11 | 94.92 |
| | Fashion-MNIST | 99.15 | 99.14 | 97.90 | 96.25 | 91.65 | 83.14 |
| | CIFAR-10 | 99.70 | 99.29 | 99.26 | 99.03 | 99.02 | 39.69 |
| AlexNet | CIFAR-100 | 99.19 | 99.15 | 99.11 | 99.16 | 99.14 | 97.88 |

Table 4: Average testing accuracy in percentages of all the models over three runs over different corruption degrees (indicated in the last six columns).

| Model | Dataset | 0% | 20% | 40% | 60% | 80% | 100% |
|---|---|---|---|---|---|---|---|
| MLP | MNIST | 97.87 | 87.38 | 73.28 | 54.16 | 32.09 | 9.81 |
| | CIFAR-10 | 56.37 | 48.62 | 40.35 | 30.55 | 19.68 | 9.80 |
| CNN | MNIST | 99.15 | 87.51 | 69.44 | 47.10 | 28.30 | 9.85 |
| | Fashion-MNIST | 90.74 | 77.74 | 61.35 | 43.26 | 25.57 | 10.08 |
| | CIFAR-10 | 74.95 | 60.48 | 46.15 | 30.96 | 18.32 | 9.89 |
| AlexNet | CIFAR-100 | 36.75 | 28.44 | 20.53 | 9.64 | 3.43 | 0.96 |

