# OpenReview forum: "Extracting Latent Generalization from Models Trained with Noisy Labels"
_NeurIPS.cc/2025/Workshop/Reliable_ML — NeurIPS 2025 - Reliable ML Workshop_

### Official Review · Reviewer_Gxjm · 2025-09-15
**Solid Empirical Findings, Missing Theory and Applications**

**Rating:** 7
**Confidence:** 3

**Review:**

### **Summary**
This paper investigates the phenomenon of “latent generalization” in models trained with noisy labels. The authors propose the Minimum Angle Subspace Classifier (MASC), which builds class-conditioned subspaces from corrupted training data (via PCA) and classifies test samples based on minimum angle distance. Surprisingly, MASC often outperforms the original model itself in test accuracy, even when label corruption is severe. This suggests that deep networks, despite memorizing noisy labels, still encode internal representations that retain substantial generalization ability. The paper provides extensive empirical evidence across multiple models (MLP, CNN, AlexNet) and datasets (MNIST, Fashion-MNIST, CIFAR-10, CIFAR-100).

### **Strength**
* Novelty: The discovery that internal representations can retain generalization despite noisy-label training is surprising and valuable.
* Rigor and Empirical Quality: Experiments span multiple datasets, architectures, and corruption levels, with careful comparisons across layers. The analysis of performance differences between models and probes is thorough.
* Clear Methodology: The construction of MASC and the experimental setup are clearly explained, making the study easy to follow.
* Relevance: The work directly addresses reliability challenges in ML under imperfect (noisy) data conditions, falling under the theme: "Learning with missing or biased data; and truncated statistics".

### **Weaknesses / Limitations**
* Lack of Theoretical Explanation: The paper does not provide a theoretical account of why latent generalization persists, especially in later layers, nor why subspace geometry resists memorization effects.
* Limited Scope of Evaluation: Experiments are restricted to image classification datasets and standard models. It is unclear if the findings extend to more complex architectures (e.g., transformers) or non-vision domains.
* Baseline Comparisons: Comparisons with alternative noise-robust training techniques (e.g., bootstrapping, loss correction, or semi-supervised methods) are missing. This makes it hard to gauge whether MASC is competitive or just an interesting phenomenon.
* Statistical Rigor: Results are averaged over only three runs; no statistical significance testing is reported.

### **Suggestions for Authors**
* Add theoretical exploration: Provide hypotheses or partial theoretical grounding for why MASC outperforms the model
* Explore practical utility, e.g., could MASC be used in real-world noisy-label scenarios to improve downstream performance?
* Broaden experiments: Test MASC on transformers or language datasets to assess generality beyond vision benchmarks.
* Stronger baselines: Compare against established label-noise-robust methods (e.g., Generalized Cross Entropy, bootstrapping, etc.)

---

### Official Review · Reviewer_CTXr · 2025-09-17
**A very insightful paper**

**Rating:** 8
**Confidence:** 3

**Review:**

Summary
This paper investigates the question of whether the latent geometric representations learned by overparameterized deep neural networks can provide stronger generalization than the networks’ final predictions, particularly in the presence of corrupted data. The authors are motivated by the observation that deep networks often memorize noise or spurious patterns when trained on imperfect datasets, which can obscure the true generalization potential of the learned internal representations. Through a series of empirical studies, they show that in many cases, these latent geometric representations retain predictive information about the ground-truth labels that is not fully captured by the model’s direct outputs. This phenomenon suggests that the internal representations may embody a form of “hidden” generalizability that conventional evaluation fails to reveal. The paper thus raises an intriguing point: overparameterized models might be capable of learning useful structures that are masked by their tendency to overfit, and careful probing of their intermediate layers can surface this potential.

Strengths
One of the key strengths of the paper lies in its framing of an established idea—probing intermediate representations of deep networks—within the novel context of memorization and generalization under data corruption. While representation probing itself has been widely studied, the authors’ application of this technique to highlight memorization dynamics offers a fresh angle and a valuable contribution. The experimental results are presented clearly and are compelling: they consistently indicate that the geometric structure of the latent space contains predictive signals that are more robust than the final outputs of the network. This empirical evidence is not only convincing on its own but also provides deep insights into the limitations of how generalization is typically measured. In doing so, the paper highlights that the predictive capacity of deep networks may often be underappreciated when one looks only at their raw accuracy on corrupted datasets. This insight opens up promising research directions, particularly for designing new evaluation methods and training strategies that better harness the generalizability embedded in latent representations. Moreover, the focus on reliability in imperfect-data scenarios makes the work especially timely and relevant to both the machine learning theory community and practitioners dealing with noisy real-world data.

Weaknesses and Limitations
The paper does not present major weaknesses or limitations. That said, one potential area that could be expanded in future work is the exploration of how these findings might translate to a broader set of architectures and tasks. For example, it remains somewhat unclear whether the observed phenomenon is equally strong across different domains (e.g., vision versus language) or whether it depends on particular training regimes. While this does not detract from the contribution of the paper, it is a natural next step for extending the impact of the results.

Suggestions for the Authors
Although the authors have already provided insightful discussion, further elaboration on practical implications would be welcome. For instance, could one design training methods that explicitly encourage the model to surface or preserve the generalizable structure in latent spaces, thereby reducing reliance on potentially overfitted final layers? Additionally, it would be interesting to explore whether existing regularization or representation learning techniques (such as contrastive learning) align with or differ from the behaviors highlighted in this study. Such connections could enrich the paper’s influence and give the community concrete ideas for follow-up work.

Ethics
No ethical concerns are apparent in this paper. The study is largely methodological and empirical, and it does not raise sensitive issues such as fairness, safety, or deployment risks.